# Non-Cardiac Chest Pain as a Persistent Physical Symptom: Psychological Distress and Workability

**DOI:** 10.3390/ijerph20032521

**Published:** 2023-01-31

**Authors:** Sigrún Ólafsdóttir Flóvenz, Paul Salkovskis, Erla Svansdóttir, Hróbjartur Darri Karlsson, Karl Andersen, Jón Friðrik Sigurðsson

**Affiliations:** 1Department of Psychology, Reykjavik University, 101 Reykjavik, Iceland; 2Oxford Centre for Psychological Health, Oxford Institute of Clinical Psychology Training and Research Oxford Cognitive Therapy Centre, Warneford Hospital, Oxford University, Oxford OX3 7JX, UK; 3The National Hospital of Iceland, 101 Reykjavik, Iceland; 4Faculty of Psychology, School of Health Sciences, University of Iceland, 101 Reykjavik, Iceland; 5Spitalzentrum Biel, 2501 Biel, Switzerland; 6Faculty of Medicine, School of Health Sciences, University of Iceland, 102 Reykjavik, Iceland

**Keywords:** PPS (Persistent Physical Symptoms), non-cardiac chest pain, NCCP, workability, medically unexplained symptoms, MUS

## Abstract

Non-Cardiac Chest Pain (NCCP) is persistent chest pain in the absence of identifiable cardiac pathology. Some NCCP cases meet criteria for Persistent Physical Symptoms (PPS), where the symptoms are both persistent and distressing/disabling. This study aimed to identify patients that might need specialist treatment for PPS by examining cases of NCCP that meet PPS criteria. We analysed data from 285 chest pain patients that had received an NCCP diagnosis after attending an emergency cardiac department. We compared NCCP patients who did and did not meet the additional criteria for heart-related PPS and hypothesised that the groups would differ in terms of psychological variables and workability. We determined that NCCP patients who meet PPS criteria were more likely than other NCCP patients to be inactive or unable to work, reported more general anxiety and anxiety about their health, were more depressed, ruminated more, and, importantly, had a higher number of other PPS. A high proportion of NCCP patients meet PPS criteria, and they are similar to other PPS patients in terms of comorbidity and disability. This highlights the importance of focusing psychological interventions for this subgroup on the interplay between the range of physical and psychological symptoms present.

## 1. Introduction

Non-Cardiac Chest Pain (NCCP) is defined as persistent chest pain in the absence of any identifiable cardiac pathology. NCCP can be caused and exacerbated by a variety of factors. Gastrointestinal, musculoskeletal, psychiatric and pulmonary factors are thought to explain a substantial proportion of NCCP cases [1,2,3], and it has been suggested that an NCCP diagnosis should only be assigned after excluding such causes. Nonetheless, the clinical care tends to mainly focus on excluding cardiac conditions and disease; an NCCP diagnosis is assigned by exclusion of such causes (although other common causes may not have been). This makes NCCP a heterogeneous “diagnosis by exclusion”, and thus, to some extent, arbitrary as the diagnosis depends on the depth to which the service provider decides to investigate.

NCCP is common across clinical settings and has a lifetime prevalence from 20 to 33% [2,4]. Chest pain is considered to be of non-cardiac origin in 70–80% of patients with chest pain in primary health care and 50–80% of such cases in Emergency Departments (ED) and rapid chest pain clinics [1,5,6,7,8]. Approximately 2–5% of all ED visits are due to NCCP, which makes it one of the most common reasons for an ED visit [2]. No gender differences have been noted in the prevalence of NCCP, and it is relatively evenly spread across all age groups.

NCCP is associated with diminished quality of life, psychological distress, reduced ability to work, high health care consumption and high societal cost. Anxiety and mood disorders commonly co-occur with NCCP [1,9]. Anxiety and depression levels of NCCP patients are similar to those with Cardiac Chest Pain (CCP) and higher than of healthy controls [10,11]. Quality of life of NCCP patients tends to be similar or slightly better than the quality of life of those with CCP, but lower than that of healthy controls [11,12,13]. Unemployment and work absenteeism rates are high among NCCP patients, higher than among CCP patients [14]. Despite a generally good long-term prognosis in terms of mortality, many NCCP patients continue to have problems with chest pain and continue to seek health care for their symptoms [6,13,14,15]. This is a burden to the health care system that leads to high health care cost and high indirect cost [2] without benefit to patients’ quality of life.

As a diagnosis, NCCP falls under the wider category of Persistent Physical Symptoms (PPS) (sometimes referred to as Medically unexplained symptoms or Functional symptoms). PPS are generally defined as persistent physical symptoms that do not have a clear physical underpinning but have a significant adverse impact on the person in terms of distress or disability. These symptoms can vary in duration and severity, from a single mild symptom to multiple severe and disabling symptoms. PPS are common in all medical settings [16,17,18,19,20] and many different types have been described and defined as syndromes (e.g., Chronic fatigue, Fibromyalgia and Irritable bowel syndrome).

PPS have been widely studied and have been linked to psychological distress [21,22,23,24], disability [22,23,25,26,27,28], work absence [22,26,27,28], high unemployment rates [29,30], risk of becoming unemployed [27,28], and needing permanent disability pension [23]. PPS are related to high health care use and costs which tends to increase with the number of PPS symptoms and symptom severity [19,31,32,33]. Medical interventions do not seem to lead to symptom improvement or reduction in distress caused by PPS, but NCCP and other specific types of PPS can be treated with approaches such as cognitive behavioural therapy (CBT) [34,35,36,37,38,39,40].

Previous studies on NCCP have relied on the absence of medically identified cardiac pathology to define the groups in question. Salkovskis, 1992 [41] suggested that a diagnosis by exclusion was not appropriate if treatment was to be considered, and that clear identification of the psychological issues was needed. By applying the definition of PPS to a group of chest pain patients whose symptoms fall under the definition of NCCP, we may be able to identify a smaller group of people whose symptoms are or are likely to become chronic and might benefit from specific psychological treatment that targets both the range of physical and psychological symptoms present and the interplay between them. In the present study, the unique contribution is the use of identification of PPS as a way of dividing patients with NCCP into those with and without detectable psychological issues. In order to identify these patients, we looked at a group of patients diagnosed with NCCP after a visit to an emergency cardiac department (CED) because no cardiac pathology could be identified, and directly compared patients who met criteria for heart-related PPS to those who did not. We hypothesise that NCCP patients that meet criteria for heart-related PPS will differ from those that do not, and that they (1) are more likely to be inactive or unable to work (2) experience more anxiety, particularly about their health, (3) ruminate more, (4) have a higher number of other PPS and (5) are more likely to be anxious and depressed.

## 2. Materials and Methods

### 2.1. Participants

This study is a part of a larger study on chest pain patients that attended the emergency cardiac department (CED) at Landspitali—The National University Hospital of Iceland between October 2015 and December 2016. Eligible patients were those who attended the CED due to chest pain, were between 18 and 65 years old, spoke Icelandic and were physically and mentally able to fill out the study questionnaires (e.g., not unconscious, intoxicated, or in critical medical condition). To be included in the current study, patients had to have received an NCCP diagnosis and have filled out the Persistent Physical Symptom Checklist (PPSC) as the list was used to divide participants into groups. Patients diagnosed with non-cardiac conditions that could explain their chest pain were excluded. Figure 1 shows the number of patients that attended the CED during the research period, eligible patients, those invited to participate, those who accepted participation and met the inclusion criteria.

Many eligible patients were not asked to participate because the CED staff did not always, at busy times, have the opportunity to do so. Missed opportunities for patient recruitment are considered random. Participants excluded because of unavailable PPSC scores were not significantly different from the remaining patients in terms of gender, age, education or marital status.

Eighty-five patients (29.8%) met criteria for heart-related PPS on the PPSC, and 200 (70.2%) did not. The NCCP patients’ age ranged from 18 to 66 years. NCCP patients that met the heart-related PPS criteria were less likely to be in a relationship and to have a university degree than those who did not. Information about gender distribution, age, education level and relationship status of both groups is presented in Table 1.

### 2.2. Measures

#### 2.2.1. Psychological Measurements

The Persistent Physical Symptom Checklist (PPSC) is a self-report screening instrument that measures persistent problems with seven types of physical symptoms, i.e., sleep problems, pain, chronic tiredness fatigue or muscle problems, gastrointestinal problems, heart and chest symptoms, dizziness and/-or related problems and gynaecological problems. Respondents were asked whether they have had problems with a particular symptom for more than six months (one month for sleep problems), whether their symptoms have a known cause, and if so, what is the cause for their problems. The reported causes were considered and coded according to whether they included a possible medical explanation for their symptoms. They were then asked to rate on a 9-point scale the extent to which their problems interfere with their lives. The criteria for a particular PPS are considered met if the problem has been present for more than six months, did not have a clear defined medical cause and was definitely interfering with the respondent’s life (≥4 points). An evaluation of the checklist in an internal sample of 55 participants showed adequate convergent validity.

The Short Health Anxiety Inventory (SHAI) [42,43] measures health-related anxiety. A cut-off score of 18 has been shown to reliably identify people meeting the DSM-IV hypochondriasis criteria. The scale has good psychometric properties [43,44], and in this sample, Cronbach’s alpha was 0.88.

The Patient Health Questionnaire-9 (PHQ-9) [45,46] measures depression severity over a two week period [47]. It has cut-off points of 5, 10, 15 and 20 (interpreted as mild, moderate, moderately severe and severe depression). The scale has good psychometric properties [45,46], and in this sample, Cronbach’s alpha was 0.71.

The Generalised Anxiety Disorder-7 (GAD-7) [48,49] measures general anxiety. It has cut-off points at 5, 10 and 15 (interpreted as mild, moderate, and severe anxiety). The scale has good psychometric properties [48,49], and in our sample, Cronbach’s alpha was 0.92.

The Perseverative Thinking Questionnaire (PTQ) [50] measures self-reported rumination. The scale has 15 items, a total score that ranges from 0 to 75 and three subscale scores. Only the total score was used in this study. The original PTQ has good psychometric properties [50] and Cronbach’s alpha was 0.97 in our sample.

#### 2.2.2. Medical Diagnosis

Diagnosis given at the CED as well as any prior cardiac diagnoses were retrieved from the patients’ medical records. The following ICD-10 diagnoses were counted as CCP: Unstable angina (I20.0), Myocardial infarction (I21), Cardiac arrest (I46), Chronic ischemic heart disease (I25), Heart failure (I50). When no diagnosis was specified in the records, an assessment was performed based on physicians’ notes. If they included information about known cardiac conditions, the patient was considered as having CCP. The following ICD-10 diagnoses were counted as NCCP: chest pain, unspecified (R07.4), other chest pain (R07.3), observation for other suspected cardiovascular diseases (Z03.5) and other diagnoses suggesting persistent physical or stress-related causes such as myositis, unspecified (M60.9), gastro-esophageal reflux disease without esophagitis (K21.9), myalgia (M79.1), acute stress reaction (F43.0) and hyperventilation (R06.4). Patients were excluded from further analysis if they had been assigned a diagnosis of an acute physical condition that could have explained the chest pain such as cholelithiasis (K80), arrhythmia (I40), pulmonary embolism (I26), aortic aneurysm and dissection (I71), cholecystitis (K81), appendicitis (K37), acute pancreatitis (K85), pericarditis (I30), and herpes zoster (B02).

#### 2.2.3. Background Information

Patients provided information about their gender, marital status, employment status and the highest completed education on a questionnaire containing multiple-choice questions. We categorised employment status in two ways. The first group was based on whether they were active, i.e., working, studying, managing a household or retired due to old age, or inactive, i.e., were unemployed, receiving rehabilitation benefits or permanent disability pension. The second group was based on whether they were able to work or not, i.e., whether they were receiving rehabilitation benefits or permanent disability pension.

### 2.3. Procedure

Nurses presented the study to eligible patients at the CED at Landspitali. They asked for the patients’ informed consent and provided self-report questionnaires to those agreeing to participate. While waiting to be seen by a doctor, participants signed an informed consent form, filled out the questionnaires and returned them to the department staff upon completion. The National Bioethics Committee of Iceland approved the study (application no. VSN-15–121).

### 2.4. Data Analysis

We divided a group of patients that had received an NCCP diagnosis after a visit to a CED in into two groups based on whether their heart and chest symptoms met PPS criteria on the PPSC. T-tests were performed to determine whether there were any significant differences in the total number of non-heart-related PPS, health anxiety scores, rumination scores, general anxiety and depression scores between the groups. To detect whether there was a significant association between having heart-related PPS and being inactive, unable to work or scoring above clinical cut off for health anxiety, general anxiety and depression, χ^2^ tests were used, and odd ratios were calculated.

## 3. Results

The first four hypotheses of our study were supported as NCCP patients whose symptoms met PPS criteria were more likely than other patients to be inactive or unable to work; they experienced more anxiety and anxiety about their health, ruminated more and had a higher number of other PPS. The fifth hypothesis was partially supported as these patients were more likely to have depressive and health anxiety symptoms, but not general anxiety symptoms, in the clinical range. As can be seen in Table 2, NCCP patients that met the criteria for heart-related PPS were more likely to be inactive or unable to work, had, on average, more general anxiety and anxiety about their health, were more depressed, and ruminated more than other NCCP patients. There was a significant association between meeting the PPS criteria and having health anxiety and depression scores in the clinical range. NCCP patients meeting these criteria were almost four times more likely to have health anxiety scores and more than five times more likely to have depression scores in the clinical range than other patients.

NCCP patients whose chest pain met PPS criteria had six times higher odds (95% CI = Y–Z) of meeting criteria for any other PPS subtype (83.5%) compared to patients whose chest pain did not meet PPS criteria (56.4%), χ^2^(1) = 34.315, *p* < 0.001. The average number of non-heart-related PPS was also significantly higher in NCCP patients that met the additional PPS criteria than in other NCCP patients. As can be seen in Figure 2, NCCP patients in general and those of them that do not meet the additional criteria most commonly do not meet criteria for any other PPS. In these groups, the number of patients meeting criteria decreases as the total number of PPS symptoms increases, i.e., it is generally more common to have no or few PPS than many. On the other hand, most NCCP patients that met the additional criteria also met criteria for one or more non-heart-related PPS. In fact, they most commonly met criteria for three to four other PPS as well.

## 4. Discussion

NCCP is defined as persistent chest pain in the absence of any identifiable cardiac pathology and falls by definition under the broader term of PPS which have been defined as persistent and disabling physical symptoms that have no known biological cause. In clinical practice, NCCP diagnoses are often assigned to chest pain patients solely because no cardiac causes were determined for their pain which results in a patient group that is large and heterogenous. We proposed that by applying PPS criteria to this patient group, a smaller group could be identified that would be similar to other PPS patients and would therefore be likely to benefit from psychological treatment specifically targeting the various physical and psychological symptoms they experience and the interplay between them. We compared NCCP patients whose chest pain did meet PPS criteria to those whose chest pain did not and, as hypothesised, detected differences in terms of ability to work, psychological symptoms and the prevalence of multiple PPS.

The two-group comparison showed elevations in all continuous measures of psychological distress in patients whose symptoms met PPS criteria with medium effect sizes. Examining clinical cut-off points for these measures revealed that NCCP patients whose symptoms met PPS criteria were almost four times more likely to have clinical levels of health anxiety and five times more likely to have depressive symptoms in the clinical range than patients whose chest pain did not meet this criteria. They were also about three times as likely to be unable to work or be inactive. This is consistent with what is already known about PPS as they have been associated with reduced quality of life, impaired functional ability, work disability [22,23,25,26,27,28,29,30] and high rates of comorbid mood and anxiety disorders [22].

NCCP patients whose chest pain met PPS criteria were six times more likely to meet criteria for any other PPS subtype compared to patients whose chest pain did not and met criteria for more PPS on average. Patients whose chest pain met PPS criteria most commonly met criteria for three to four other PPS and only a minority of them (16.5%) did not meet criteria for any other PPS. In contrast, patients whose chest pain did not meet PPS criteria most commonly did not meet criteria for any other PPS, and if they did, they rarely met criteria for more than two PPS (10%). These results align with previous studies that show that when the patients’ symptoms meet criteria for a specific PPS, they often meet criteria for more than one such condition [51,52,53], which corroborates the clinical importance of multiple PPS.

Our study has some limitations that might affect the generalizability of the results. The overall sample of 285 NCCP patients was comparable to previously studied samples in terms of age, gender, educational level and marital status and symptoms of anxiety and depression were broadly similar to what has previously been reported in similar settings [11,12,14,15,54]. The average levels of depressive symptoms might, however, be slightly higher and the proportion of participants with anxiety problems slightly smaller than in previous studies [11,15]. This difference is possibly explained by the fact that different scales and cut-off scores have been used in different studies, but it is also possible that cultural factors (including differences in healthcare systems) might explain this. The cultural characteristics of the population might also have influenced the results, although this possible limitation of generalisability is not likely to be of major concern as Icelanders tend to be similar to other North European populations in terms of reported psychological characteristics. Another possible limitation is that not all eligible chest pain patients were invited to participate in the study. The missing recruitment opportunities are considered random with regard to the associations under study as they were caused by the CED staff not having the opportunity to present the study to their patients at busy times.

## 5. Conclusions

Formulating patients’ problems accurately is important in selecting appropriate treatment. General medical treatment does not seem to lead to symptom improvement in the case of PPS, while CBT has been shown to be effective for different PPS subtypes, including NCCP. Referring all patients with an NCCP diagnosis to psychological assessment or CBT might, however, not be appropriate or feasible as the group is both heterogenous and large. Our results show that a significant proportion (almost 30%) of people diagnosed with NCCP have symptoms that meet PPS criteria while the majority of patients do not. Among patients that receive NCCP diagnosis, there seems to be a subgroup of patients where PPS is an accurate formulation of their problem. Most of these patients seem to be experiencing the more complex problem of multiple PPS and many also have significant symptoms of depression and/or health anxiety. This raises the exciting possibility of focusing effective psychological treatment on this subgroup which could benefit those most seriously affected and most likely to respond to such treatment [41]. In order to be able to refer people from the large and heterogeneous group of NCCP patients to appropriate assessment and treatment, we need to directly identify those patients whose symptoms are best formulated as a PPS or multiple PPS. Our results suggest that this can perhaps be achieved in a relatively simple and cheap manner with self-report, although further studies are needed to determine a way in which this can be achieved most accurately.

## Figures and Tables

**Figure 1 ijerph-20-02521-f001:**
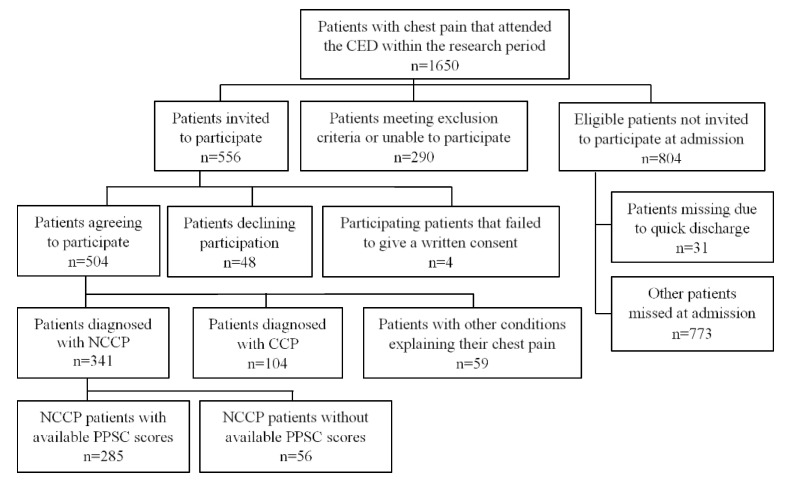
Flow chart of participant selection from patients attending the CED. Note. CED = Cardiac emergency department; NCCP = Non cardiac chest pain; CCP = Cardiac Chest Pain; PPSC = Persistent Physical Symptom Checklist.

**Figure 2 ijerph-20-02521-f002:**
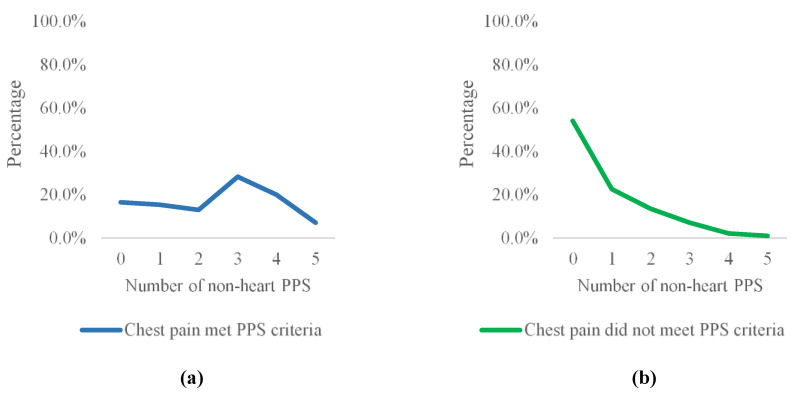
(**a**) Percentage of participants whose chest pain symptoms met PPS criteria and who met criteria for none, one or more non-heart-related PPS. (**b**) Percentage of participants whose chest pain did not meet PPS criteria and who met criteria for none, one or more non-heart-related PPS.

**Table 1 ijerph-20-02521-t001:** Gender, education and relationship status of NCCP patients.

Demographic Variables	NCCP Patients	NCCP Patients with Heart Related PPS	NCCP Patients without Heart Related PPS	χ^2^
Gender				
Male	154 (54.0%)	48 (56.5%)	106 (53.0%)	0.29
Female	131 (46.0%)	37 (43.5%)	94 (47.0%)	
Education completed				
Primary school	60 (21.8%)	24 (28.9%)	36 (18.8%)	7.86 *
Secondary school	102 (37.1%)	35 (42.2%)	67 (34.9%)	
University degree	113 (41.1%)	24 (28.9%)	89 (46.4%) *	
Relationship status				
Single, separated or widowed	66 (23.9%)	26 (32.1%)	40 (20.5%) *	4.22 *
Married or cohabiting	210 (76.1)	55 (67.9%)	155 (79.5%) *	
	*M (SD)*	*M (SD)*	*M (SD)*	*T*
Age	49.84 (10.68)	48.07 (12.66)	50.60 (9.65)	1.646

Note: * *p* < 0.05.

**Table 2 ijerph-20-02521-t002:** Comparisons of working status, rumination and symptoms of anxiety, depression and health anxiety between NCCP patients that did and did not meet criteria for heart-related PPS.

Measure	All NCCP Patients	Heart-Related PPS Criteria Met	Heart-Related PPS Criteria Not Met	Significance	Effect Size
	*N* (%)	*N* (%)	*N* (%)	χ^2^	*OR*
Inactivity	27 (10.1%)	14 (18.2%)	13 (6.8%)	7.751 *	3.03
Inability to work	22 (8.2%)	11 (14.3%)	11 (5.8%)	5.231 *	2.71
SHAI _≥18_	17 (6.2%)	10 (12.2%)	7 (3.6%)	7.217 *	3.67
GAD-7 _≥10_	40 (14.4)	17 (20.7%)	23 (11.8%)	3.732	1.96
PHQ-9 _≥10_	47 (17.0%)	29 (34.9%)	18 (9.3%)	26.949 *	5.22
	*M (SD)*	*M (SD)*	*M (SD)*	*t*	*r*
SHAI _Mean (SD)_	8.79 (5.52)	10.88 (6.83)	7.90 (4.60)	−3.615 *	0.32
GAD-7 _Mean (SD)_	4.70 (5.02)	6.77 (5.81)	3.83 (4.38)	−4.112 *	0.35
PHQ-9 _Mean (SD)_	6.50 (3.98)	8.47 (4.57)	5.65 (3.38)	−5.059 *	0.42
PTQ _Mean (SD)_	14.54 (12.44)	18.68 (13.68)	12.78 (11.46)	−3.427 *	0.29
Total number of non-heart-related PPS	1.31 (1.46)	2.41 (1.55)	0.84 (1.13)	−8.493 *	0.61

Note. NCCP = Non-cardiac chest pain; PPS = Persistent physical symptoms; OR = Odds ratio; SHAI = Short Health Anxiety Inventory; PHQ-9 = Patient Health Questionnaire-9; GAD-7 = Generalised Anxiety Disorder-7; PTQ = Perseverative Thinking Questionnaire. * *p* < 0.05.

## Data Availability

The data that support the findings of this study can be made available on request from the corresponding author. The data are not publicly available due to privacy or ethical restrictions.

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
