# Peer review of "Non-Cardiac Chest Pain as a Persistent Physical Symptom: Psychological Distress and Workability"

_ijerph, 2023, doi:10.3390/ijerph20032521_

Round 1
Author Response
"Please see the attachment.

Reviewer 2 Report
I thank the authors for sharing their interesting manuscript. I would like to see the following minor revisions:
1. The authors formulate the aim of the study in the abstract, but omit it in the introduction. It seems to me appropriate to add the aim of the study in the introduction.
2. The authors formulate hypotheses of the study, but then in the discussion they do not indicate whether these hypotheses were confirmed or disproved. I would add at least a brief statement to the results or discussion (e.g., "Hypothesis 1 is fully confirmed").
3. There is no reference to the Persistent Physical Symptom Checklist, so it is unclear whether the authors used some valid and reliable instrument or developed their own. In the first case, the source should be stated, and in the second case, the instrument, its development, and preliminary psychometric properties should be described in detail.
4. Line 207: missing dot and space between «p < 0.001» and «The average number…».
5. I would recommend adding the limitations of the study, its prospects and practical implications.
Author Response
"Please see the attachment

Round 2
Reviewer 1 Report
The current revision is accepted